# An Extension of the Bland–Altman Plot for Analyzing the Agreement of More than Two Raters

**DOI:** 10.3390/diagnostics11010054

**Published:** 2021-01-01

**Authors:** Sören Möller, Birgit Debrabant, Ulrich Halekoh, Andreas Kristian Petersen, Oke Gerke

**Affiliations:** 1Department of Clinical Research, University of Southern Denmark, 5000 Odense C, Denmark; moeller@health.sdu.dk (S.M.); Oke.Gerke@rsyd.dk (O.G.); 2Open Patient data Explorative Network, Odense University Hospital, 5000 Odense C, Denmark; 3Department of Public Health, Epidemiology, Biostatistics and Biodemography, University of Southern Denmark, 5000 Odense C, Denmark; bdebrabant@health.sdu.dk (B.D.); uhalekoh@health.sdu.dk (U.H.); 4Department of Research and Learning, Hospital of Southern Jutland, 6200 Aabenraa, Denmark; Andreas.Kristian.Pedersen@rsyd.dk; 5Department of Nuclear Medicine, Odense University Hospital, 5000 Odense C, Denmark

**Keywords:** Bland–Altman plot, agreement, visualization, simulation study, method comparison, inter-rater, intra-rater

## Abstract

The Bland–Altman plot is the most common method to analyze and visualize agreement between raters or methods of quantitative outcomes in health research. While very useful for studies with two raters, a limitation of the classical Bland–Altman plot is that it is specifically used for studies with two raters. We propose an extension of the Bland–Altman plot suitable for more than two raters and derive the approximate limits of agreement with 95% confidence intervals. We validated the suggested limit of agreement by a simulation study. Moreover, we offer suggestions on how to present bias, heterogeneity among raters, as well as the uncertainty of the limits of agreement. The resulting plot could be utilized to investigate and present agreement in studies with more than two raters.

## 1. Introduction

In health research, it is often desired to determine to what degree different raters or methods (either persons or devices) agree on the measurement of a continuous outcome on the same subject (patient, diagnostic image, biological sample, etc.). The main aim is to ensure that the variability between raters is small enough to use observations from a single rater in future studies or clinical practice and that it does not make any difference which rater conducted the single measurement.

Typically this situation will be dealt with by applying a mixed effect regression model or related methods [1,2,3]. In addition to these quantitative results, a clinical researcher often wishes to present the agreement graphically, to ease the interpretation and communication of the results. The most common method used for this aim is the Bland–Altman (BA) plot [4], which plots differences between two raters against respective means together with 95% limits of agreement (LOAs). The interpretation and reporting of these LOAs regularly results in confusion; see [5,6,7] for recent overviews.

One of the drawbacks of the classical Bland–Altman plot is that it only applies to a situation with two raters or methods, while for both practical reasons (distributing workload to multiple raters) and statistical reasons (increased power and strengthened generalizability), it is desirable to use more than two raters for carrying out agreement studies. The standard suggestion is usually to present multiple Bland–Altman plots for each pairwise comparison of raters [3,8], but this becomes awkward for more than four raters, as *m* raters result in m·(m+1)/2 plots, which is difficult to present and cumbersome to interpret. Moreover, the pairwise nature of the limit band in this situation restricts the interpretability of the bands.

Some suggestions include presenting all observations in the same plot, with one marker for each observation instead of each subject, either connecting observations for the same subject by lines [3] or by different marker symbols [9]. However, both of these approaches become confusing when including many observations. Other suggested approaches decrease the number of plots to m−1 under the assumption of one method being a gold standard [10] or reduce the presentation to one plot, but being much different in appearance from a classical BA plot and removing the concept of an LOA, which can be compared with prespecified levels of clinically meaningful differences [11].

The aim of this paper is to present a novel extension of the classical BA plot together with a generalized LOA, which only requires one plot and one marker per observed subject. While the main aim of this proposal is the comparison of multiple raters, the plot can be applied to comparisons of multiple methods as well. Especially for this case, it is important to be aware of the assumption of homoscedasticity, which our proposal inherits from the classical Bland–Altman plot; deviations from this assumption damage the interpretation of the Bland–Altman limits of agreement [12,13].

## 2. Materials and Methods

### 2.1. Derivation of an LOA for *m* > 2 Raters

We are interested in investigating a situation with *m* raters who observe each of *n* subjects once, resulting in m·n observations $x_{i,j}$.

In this setting, we assume a data generating process similar to the assumptions made by Bland and Altman [4] for the classical BA plot, but extended to more than 2 raters:$$\begin{array}{cc} X_{i,j} & =\mu_{i}+\gamma_{j}+\epsilon_{i,j}\;i=1,\cdots ,n,j=1,\cdots ,m. \end{array}$$

Each subject’s true value $\mu_{i}$ and each rater’s bias $\gamma_{j}$ are deterministic, and the measurement error term $\epsilon_{i,j}∼N\left(0,\sigma^{2}\right)$ determines the variation in the observations, assuming that all $\epsilon_{i,j}$ are pairwise independent with constant variance.

In the classical Bland–Altman plot (for m=2; see the lower row of Figure 1), the points:$$\begin{array}{cc} \left({\bar{x}}_{i},d_{i}\right) & \overset{def}{=}\left(\frac{x_{i,1}+x_{i,2}}{2},x_{i,1}-x_{i,2}\right),\;i=1,\cdots ,n \end{array}$$
are calculated and plotted together with a bias line at:$$\begin{array}{cc} \bar{d} & \overset{def}{=}\frac{1}{n}\sum_{i=1}^{n}d_{i}=\frac{1}{n}\sum_{i=1}^{n}x_{i,1}-\frac{1}{n}\sum_{i=1}^{n}x_{i,2} \end{array}$$
and LOA at:$$\begin{array}{c} \left(U,L\right)\overset{def}{=}\bar{d}\pm t_{0.975,n-1}\cdot \sqrt{1+\frac{1}{n}}\cdot s_{d}. \end{array}$$

Here:$$\begin{array}{c} s_{d}\overset{def}{=}\sqrt{\frac{1}{n-1}\sum_{i=1}^{n}{\left(d_{i}-\bar{d}\right)}^{2}} \end{array}$$
is the empirical standard deviation of the observed differences. Note that the LOA corresponds to a 95% prediction interval for a new Gaussian distributed difference.

Instead of $t_{0.975,n-1}\cdot \sqrt{1+\frac{1}{n}}$, approximations are applied in practice, mostly either 1.96, the large *n* limit, or the approximation 2. Usually, the term $\sqrt{1+\frac{1}{n}}$ is dropped (see [3,8] for discussions on this issue). In this paper, we apply the exact factor $t_{0.975,n-1}\cdot \sqrt{1+\frac{1}{n}}$ in the derivation of the classical BA LOA [10] due to our later investigations into small sample sizes such as n=10.

For the case of m>2 raters, we suggest plotting:$$\begin{array}{cc} \left({\bar{x}}_{i},s_{i}\right) & \overset{def}{=}\left(\frac{1}{m}\sum_{j=1}^{m}x_{i,j},\sqrt{\frac{1}{m-1}\sum_{j=1}^{m}{\left(x_{i,j}-{\bar{x}}_{i}\right)}^{2}}\right),\;i=1,\cdots ,n, \end{array}$$
resulting in an x-axis corresponding to the classical Bland–Altman plot, but with the y-axis presenting the intra-subject standard deviation instead of the observed differences, resulting in only one marker for each observed subject even with more than 2 raters. This is in accordance with a suggestion made earlier by Bland and Altman [4], but not yet widely applied. Note that this change mathematically corresponds to switching from plotting an $L_{1}$ distance on the y-axis to plotting an $L_{2}$ distance instead.

If m=2, this results in a plot corresponding to the classical Bland–Altman plot in which the points of the lower half of the BA plot are mirrored at the bias line and the *y* axis is scaled by a factor of $\sqrt{2}$. Figure 1 contrasts our suggestion with the classical Bland–Altman plot for m=2, while Figure 2 gives examples of our plot for m>2 and different choices of *n*.

While it is intuitive where the sample should be placed, the placement of a 95% LOA requires some additional considerations. Here, we use the property that the empirical variance of *m* independent, normally distributed random variables with equal mean (hence, excluding systematic bias between raters) and constant variance is $\chi^{2}$-distributed with m−1 degrees of freedom, and hence, the empirical standard deviation will be χ-distributed with m−1 degrees of freedom, ignoring appropriate scaling factors.

Applying this to our model, we observe asymptotically:$$\begin{array}{cc} \frac{s_{i}}{\sigma }\sqrt{m-1} & ∼\chi (m-1)\;i=1\ldots n,j=1\ldots m. \end{array}$$

Therefore, we propose to place our LOA at:(1)$$\begin{array}{cc} L & \overset{def}{=}\chi_{0.95,m-1}\frac{1}{\sqrt{m-1}}s \end{array}$$
where $\chi_{0.95,m-1}$ denotes the quantile function of a χ-distribution with m−1 degrees of freedom and $s=n^{-1}\sum_{i}^{n}s_{i}$ is the average intra-subject standard deviation. Investigating agreement between methods or raters, this LOA then can be compared to a clinically relevant difference by rescaling the y-axis with a factor of $\sqrt{2}$, which enables direct comparison of deviations from the different ratings as used in our method with differences of 2 ratings as used in classical BA plots. We decided not to propose this rescaling in general for the extended plots, as the rescaled y-axis loses its clear statistical interpretation as a within-subject standard deviation.

Note that this formula only takes into account the number of raters, but neither the number of subjects observed (that is the uncertainty in *s*, which in the classical Bland–Altman plot is taken into account by using the *t*-quantile), nor that the coverage achieved by the LOA should correspond to a 95% prediction interval for an additional observation instead of covering 95% of the original observations [14] (the factor $\sqrt{1+1/n}$ in the classical Bland–Altman plot); hence, our formula is only expected to be precise for large *n*. To communicate the uncertainty of the LOA, we suggest plotting 95% confidence intervals around the line. Here, we did this by applying bootstrapping in each sample with 1000 repetitions and reporting a 95% bias-corrected and accelerated confidence interval. For comparison, we plot the exact confidence intervals suggested by Carkeet [15] in the classical BA plots presented in this paper.

### 2.2. Suggestion for Indicating Bias

One limitation of our suggested plot is losing the information on a possible bias of a specific rater, as well as possible heterogeneity between the uncertainty of raters. To include information on this phenomenon in the plot, we suggest adding tick marks on the y-axis corresponding to:$$\begin{array}{cc} \left|B_{j}\right| & \overset{def}{=}\left|\frac{1}{N}\sum_{i=1}^{n}x_{i,j}-{\bar{x}}_{i}\right|\;i=1,\cdots n,j=1,\cdots m, \end{array}$$
the absolute mean difference between rater *j*’s observation and each subjects average observation. Moreover, we suggest marking each point in the plot by a color (or symbol) to distinguish which rater was responsible for the largest deviation from this subject’s mean; hence, a color/symbol very common in the plot would indicate a rater with larger uncertainty than the remaining raters, violating the assumption of σ not depending on *j*. Figure 3 shows these additions in a scenario without bias, a scenario in which Rater 2 systematically measures, on average, one standard deviation above the true value, and a scenario in which Rater 2 has much larger uncertainty than the remaining raters. Moreover, Figure 4 presents a scenario in which variability increases with the true mean, corresponding to the funnel shape appearing in a classical BA plot. The full algorithm for preparing an extended Bland–Altman plot is presented as Algorithm 1.
**Algorithm 1**Extended Bland–Altman plot for multiple raters.**Observe number of raters *m* and number of subjects *n* and individual ratings**xi,ji=1⋯n,j=1⋯m**Determine each subject’s mean and standard deviation**:(x¯i,si)=1m∑j=1mxi,j,1m−1∑j=1mxi,j−x¯i2,i=1,⋯,n,**Determine the position of the LOA**: L=χ0.95,m−11m−1swiths=n−1∑insi.**Estimate the 95% confidence interval for the LOA by bootstrapping****Determine bias indicators for each rater**:Bj=1N∑i=1nxi,j−x¯ii=1⋯n,j=1⋯m,**Plot (x¯i,si), *L* with the confidence interval, and Bj for all raters in a figure**

### 2.3. Simulation Study on Coverage

To determine how reasonably our LOA agrees with the desired nominal 95% coverage level for one new observation, we carried out a simulation. We employed 10,000 simulated samples, as this, in the worst case scenario, should result in Monte Carlo SE below 0.5% according to Morris et al. [16]. Data were generated by the process:$$\begin{array}{cc} X_{i,j} & =\mu_{i}+\epsilon_{i,j}\;i=1,\ldots ,n+1,j=1,\ldots ,m, \end{array}$$
that is without systematic bias, for each combination of m=2,3,4,5 raters and n=10,20,100 subjects. For each of these 12 combinations, we determined the coverage of the LOA based on simulated data. Furthermore, we calculated the empirical 95% quantile of points, both for the original *n* observations used to produce the plot and for one new observation (i=n+1) from the same process for each sample.

R Version 3.6.1 [17] with the packages nnet [18] and boot [19,20] was used to carry out the simulations and produce the figures. R scripts, including the seeds applied in the simulations, are available as Appendix A.

## 3. Results

### 3.1. Simulation Results

Figure 1 compares the classic and extended Bland–Altman plots in the case of two raters with simulated data. Figure 2 shows examples of generalized Bland–Altman plots for varying numbers of raters and observed subjects. Table 1 shows the empirical 95% LOA compared with the line suggested by our Formula (Equation 1), as well as the coverage obtained by applying the LOA of our formula. It can be seen that the coverage generally is between 0.92 and 0.96, although most typically slightly below 0.95. We generally see a coverage slightly above 95% and a narrower LOA than expected from the formula for the original observations. On the other hand, for the new observations, the coverage is slightly below 95%, and the empirical quantile is slightly wider than the LOA, although the obtained coverage of 93 to 95% is deemed acceptable. This decreased coverage is expected, as our formula for the LOA does not take into account the additional uncertainty of the new observations.

Figure 3 shows the indications of bias in a single observer, as well as increased uncertainty in a single rater compared to a situation without these phenomena. In the second row (with bias), it can clearly be seen that Rater 2’s tick mark is elevated compared to the other raters, while in the third row (with increased measurement error), the plot is dominated by red points, corresponding to Rater 2 most often being further away from the intra-subject mean. Hence, both deviations from model assumptions can be detected and distinguished by the extended BA plot with bias tick marks and color coded observations.

Figure 4 presents a situation in which the assumption of homogenous variance is violated, as the measurement error increases with increasing true mean. Again, we compare classic and extended Bland–Altman plots on the same data, and the inhomogeneity of variance is clearly visible as a funnel shape in both plots.

### 3.2. Application to Real World Data

To investigate the usefulness of the extended BA plot, we applied it to two datasets with multiple raters from the literature. For the first application (Figure 5 left), we used data from Wiinholt et al. [21] corresponding to measurements of tissue volume in 30 control mice by four raters. Our plot corresponds to the results presented in the left side of Figure 3 of [21] as six separate BA plots. Our plot indicates an LOA around 0.85 with a 95% up to almost one and no clear bias, but possibly a tendency of Method 4 to be more variable compared to the other methods.

For the second application, we applied the extended BA plot to the glucose measurement data from a method comparison study [22] with four methods as provided in the MethComp [23] package for R (Figure 5 right). Our plot indicates an LOA around 1.2 with a narrow confidence interval and no clear indication of either bias or heterogeneity of variance.

## 4. Discussion

### 4.1. Statement of Principal Findings

We showed that it is possible to present the agreement of more than two methods or raters in one combined plot and to obtain similar information as can be obtained from a classical BA plot including LOA and the corresponding confidence intervals; moreover, we indicated the possibility of detecting inhomogeneity in variance, as well as bias among raters. We showed by simulation that our suggested formula for the LOA of the extended BA obtains approximately 95% coverage.

### 4.2. Strengths and Weaknesses of the Study

The main strength of this study is that we combined a closed formula for the LOA based on mathematical arguments with transparent simulations documenting that the method behaves well even with small sample sizes. Moreover, we showed that our suggested plot can combine information presented in multiple plots in applications from the literature. A weakness of this study is that the LOA is only an approximation, as it does not fully capture the uncertainty of new observations, as well as only being equipped with approximate bootstrapped confidence intervals.

### 4.3. Strengths and Weaknesses in Relation to Other Studies, Particularly any Differences in the Results

Compared to the literature, the main strength of our approach is that only one plot is required for the comparison of multiple methods or raters and that only one dot is included in the plot for each observation unit, resulting in plots that are easier to comprehend. Moreover, these facts imply that the points on our plot are independent observations facilitating the estimation of accurate LOA and confidence intervals. Although it is possible to indicate bias and variance heterogeneity in our plot, it might be less clear than in some of the methods presented in the literature. Additionally, as our plot is only one-sided due to the non-negativity of standard deviations, this might impede its interpretation for readers used to classical BA plots.

An additional weakness of the proposed method is that it shares the assumption of homoscedasticity with the classical BA plot. It is well known from the literature that this assumption can be problematic, especially in the case of method comparison studies [12,13]. There have been detailed proposals of replacements of the classical BA plot, which handle heteroscedasticity in a more suitable manner both theoretically [24,25] and with practical applications and software implementations [26,27,28]. We consider this problem to be mainly prominent in the case of method comparison studies and to a lesser degree when comparing raters. As the suggested approach by Taffé (similarly to the classical BA plot) only handles comparison of two methods, we consider our proposal to be a relevant extension of the classical BA plot in the case of multiple raters and homoscedasticity.

### 4.4. Meaning of the Study: Possible Mechanisms and Implications for Clinicians or Policymakers

The possibility to present agreement information for more than two raters hopefully increases the possibility to report this information in agreement studies and to better inform the readers. Indirectly, this also might increase the inclination to include multiple raters in agreement studies, improving the generalizability of such studies. This implies that researchers can include the same observations in a graphical presentation as in more advanced analyses in studies with multiple raters, e.g., mixed models, improving the transparency of the studies. The fact that our LOA is presented with 95% confidence bands hopefully increases the likelihood that users of this plot will include confidence bands as well, which still is not the case in many papers published with classical BA plots [5,29]. One of the challenges for the extended BA plot, as well as for most other suggested extensions or replacements of the classical BA plot, is the inertia in health research of replacing commonly used methods (think, for instance, of scatter plots of two methods’ raw measurements with a 45 degree line and respective correlation coefficients back in the 1970s). We hope that our suggested plot is similar enough to the classical BA plot to aid its adoption in practice.

### 4.5. Unanswered Questions and Future Research

There are still some outstanding questions with regard to our suggested extension of the BA plot. Firstly, it would be desirable to obtain a more precise, non-asymptotic formula for the LOA taking into account the full amount of uncertainty in the observations. Secondly, a closed formula for the confidence interval would be preferable to our bootstrap approach. Finally, implementations of our plot in standard software packages will improve its applicability.

## Figures and Tables

**Figure 1 diagnostics-11-00054-f001:**
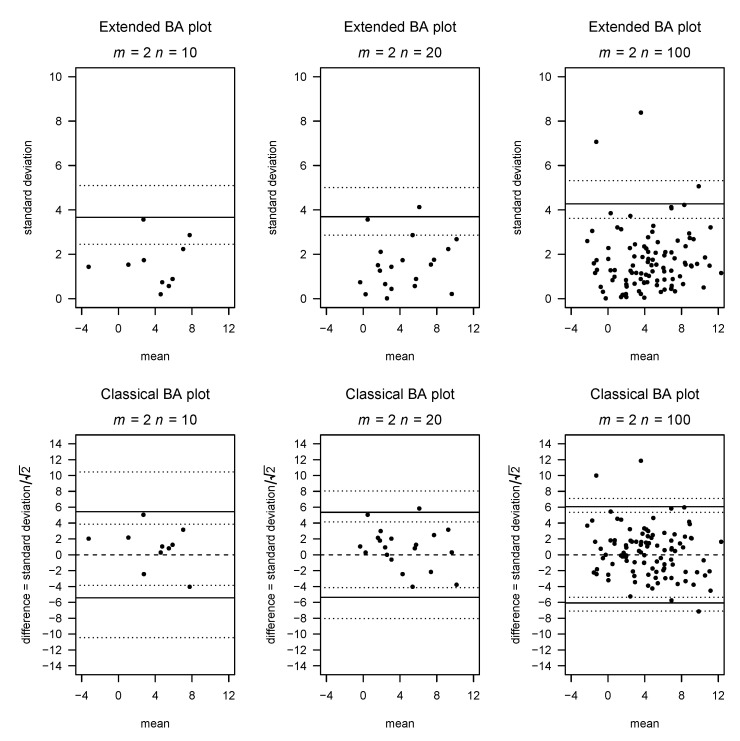
Comparison between the extended and classical BA plot for 2 raters based on simulated data with a true standard deviation σ=2.

**Figure 2 diagnostics-11-00054-f002:**
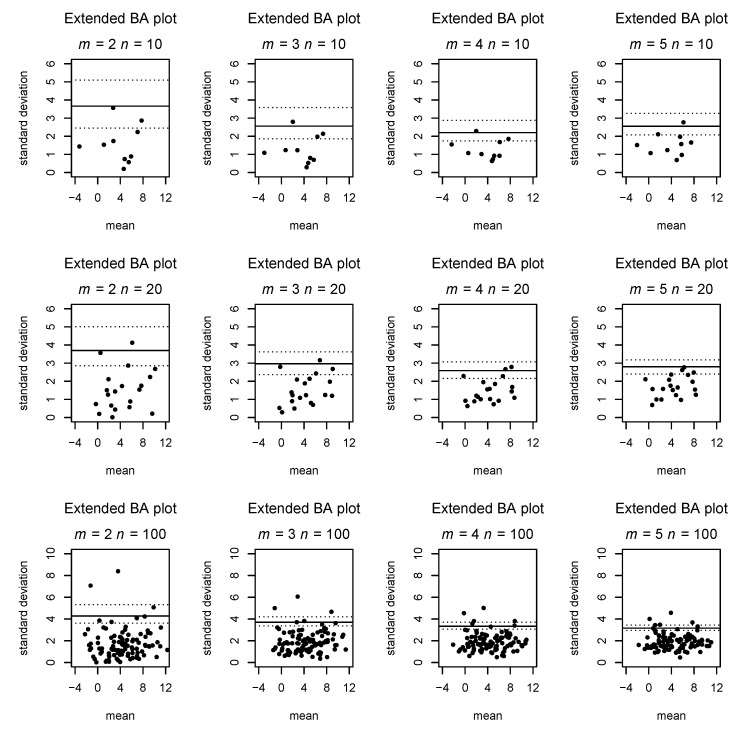
Extended BA plot for varying number of raters and observed subjects based on simulated data.

**Figure 3 diagnostics-11-00054-f003:**
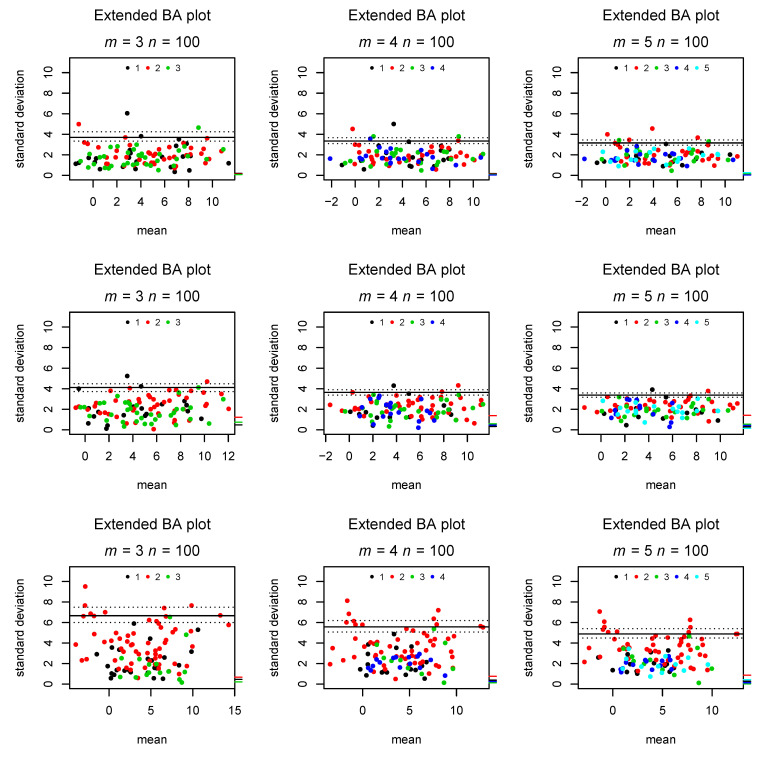
Marking the rater responsible for the largest deviation from the intra-subject mean by colored dots and the absolute rater bias by tick marks on the right y axis. The first row demonstrates absent bias and heterogeneity in error, the second row only bias, and the third row only heterogeneity ($\sigma =\sqrt{40}$ for Rater 2 in the third row; all others σ=2).

**Figure 4 diagnostics-11-00054-f004:**
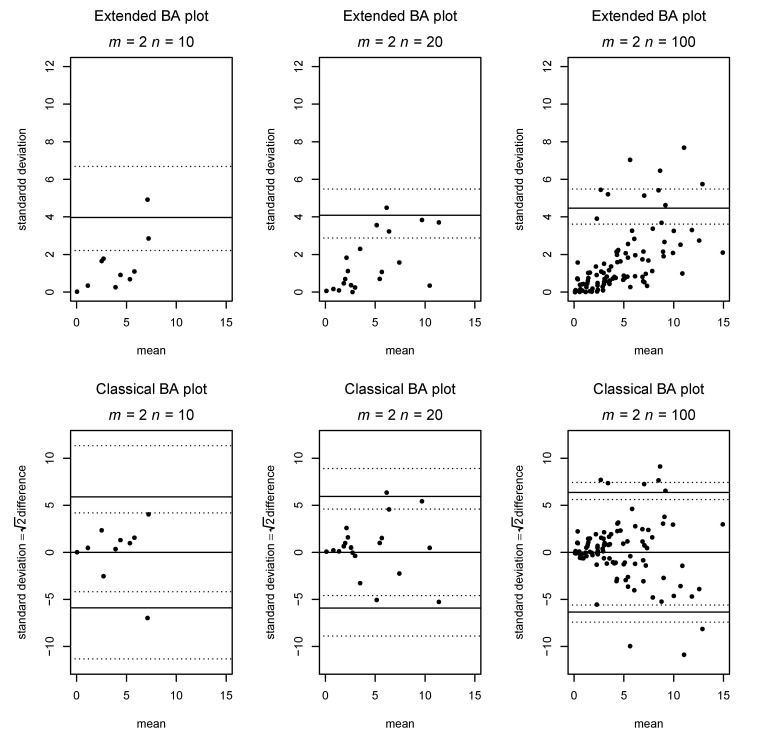
Comparison between the classic and generalized BA plot in a situation where the measurement error increases with the true value.

**Figure 5 diagnostics-11-00054-f005:**
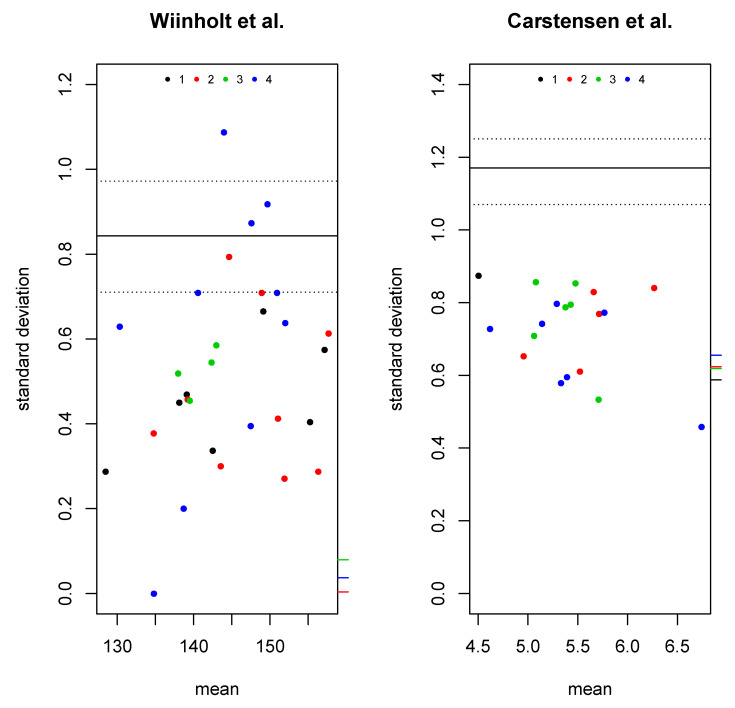
Extended BA plot on real-world data. Wiinholt et al. [21] (**left**) and Carstensen et al. [22] (**right**).

**Table 1 diagnostics-11-00054-t001:** Coverage and empirical 95% quantiles from the simulations of different choices of *m* and *n*. Results are reported both for the original observations used to estimate the LOA, as well as for the new observations obtained from the same data generating process.

		Original Observations			New Observations		
		***n*** **= 10**	***n*** **= 20**	***n*** **= 100**	***n*** **= 10**	***n*** **= 20**	***n*** **= 100**
**Raters**	**Formula** (Equation 1)	**95% Quantile**	**95% Quantile**	**95% Quantile**	**95% Quantile**	**95% Quantile**	**95% Quantile**
*m* = 2	1.959964	1.907143	1.939001	1.954703	2.188023	2.069437	1.963274
*m* = 3	1.730818	1.681992	1.709057	1.726664	1.880320	1.780882	1.741569
*m* = 4	1.613973	1.572103	1.609547	1.612125	1.706937	1.654243	1.624840
*m* = 5	1.540108	1.525253	1.536958	1.534751	1.594541	1.57577	1.552975
		Coverage	Coverage	Coverage	Coverage	Coverage	Coverage
*m* = 2		0.9575	0.9528	0.9506	0.9247	0.9400	0.9497
*m* = 3		0.9594	0.9542	0.9507	0.9253	0.9404	0.9488
*m* = 4		0.9595	0.9539	0.9510	0.9297	0.9417	0.9479
*m* = 5		0.9597	0.9541	0.9508	0.9361	0.9408	0.9462

## Data Availability

The simulated data presented in this study are reproducible by the R code in the Appendix A. The real word example data is available in publicly accessible repositories, as referenced in the Appendix A.

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
