# Peer review of "An Extension of the Bland–Altman Plot for Analyzing the Agreement of More than Two Raters"

_diagnostics, 2021, doi:10.3390/diagnostics11010054_

Round 1

Reviewer 1 Report

Interesting idea that is very well investigated and reported. I have 3 comments:

  1. The Bland-Altman method is also used for comparing 2 methods of measurement - can the proposed method be used for comparing more than 2 methods of measurement?
  2. One nice aspect of the Bland-Altman method is the LOA are simple to interpret and allow clinical experts to speculate on the clinical significance of the level of agreement between the 2 methods/raters. How would clinical experts use the proposed method to consider the clinical significance of the differences between the multiple raters?
  3. Unfortunately the simplicity of the Bland-Altman method and particularly the plot is lost with the proposed method. I expect non-statistical researchers would be reluctant to use the proposed method. 

Author Response

Interesting idea that is very well investigated and reported. I have 3 comments:

Answer: Thank you for the friendly review, we have answered your points below and amended the manuscript.

  1. The Bland-Altman method is also used for comparing 2 methods of measurement - can the proposed method be used for comparing more than 2 methods of measurement?

Answer: We have commented on this relevant point in the introduction. Our proposed method can be applied to method comparison studies, but some extra care with respect to the model assumptions is needed:

“While the main aim of this proposal is the comparison of multiple raters, the plot can be applied to comparisons of multiple methods as well. Especially in this case, it is important to be aware of the assumption of homoscedasticity, which our proposal inherits from the classic Bland-Altman plot; deviations from this assumption damage the interpretation of the Bland-Altman limits of agreement [Taffe2018, Carstensen2010b]”

  1. One nice aspect of the Bland-Altman method is the LOA are simple to interpret and allow clinical experts to speculate on the clinical significance of the level of agreement between the 2 methods/raters. How would clinical experts use the proposed method to consider the clinical significance of the differences between the multiple raters?

Answer:  We agree this is an important point and have added a description of how to interpret the y-axis and the LOA in the methods section:

“Investigating agreement between methods or raters, this LOA then can be compared to a clinically relevant difference by rescaling the y-axis with a factor of sqrt(2), which enables direct comparison of deviations from the different ratings as used in our method with differences of two ratings as used in classical BA plots. We have decided not to propose this rescaling in general for the extended plots, as the rescaled y-axis loses its clear statistical interpretation as an within-subject standard deviation.”

  1. Unfortunately the simplicity of the Bland-Altman method and particularly the plot is lost with the proposed method. I expect non-statistical researchers would be reluctant to use the proposed method. 

Answer: We agree that this is an important challenge for most newly proposed methods. We have added this point to the discussion section:

“One of the challenges for the extended BA plot, as well as for most other suggested extensions or replacements of the classical BA plot, is the inertia in health research of replacing commonly used methods (think, for instance, of scatter plots of two methods’ raw measurements with a 45 degree line and respective correlation coefficients back in the 1970s). We hope that our suggested plot is similar enough to the classical BA plot to aid its adoption in practice.”

Reviewer 2 Report

The manuscript describes a generalization to the Bland-Altman plot for the number of raters (2 or more). The question is of general utility. The suggested method is intuitively correct.

The thorough presentation, and mathematical treatment of the 95% confidence interval, attention to related questions of inter-rater bias, variance, and error trends makes this a very useful paper beyond the description of an intuitive method. It was a pleasure to read the complete treatment of the potential problems and their solutions.

My primary suggestion is to improve the presentation of the basic algorithm to present the details of the LOA and 95% confidence interval along with the basic transformation in one place. Specifically, the equation for \bar{x}_i, s_i on top of page 3, the formula for L on the bottom of page 4, and the description of the bootstrap method for 95% CI of LOA can be repeated in one table or algorithm description succinctly. This way the reader can quickly understand the complete method in one place.

Also, one would want to know, for figures 1 and 2, the values of s or sigma so the numerical values of the y-axis can be made sense of. I assumed this was 1, but it is disorienting to try to make assumptions while trying to understand someone's method. Similarly for completeness sake, the parameter values for Figure 3 can also be given as a table.

Author Response

The manuscript describes a generalization to the Bland-Altman plot for the number of raters (2 or more). The question is of general utility. The suggested method is intuitively correct.

The thorough presentation, and mathematical treatment of the 95% confidence interval, attention to related questions of inter-rater bias, variance, and error trends makes this a very useful paper beyond the description of an intuitive method. It was a pleasure to read the complete treatment of the potential problems and their solutions.

Answer: Thank you for the friendly review, we have answered your points below and amended the manuscript.

My primary suggestion is to improve the presentation of the basic algorithm to present the details of the LOA and 95% confidence interval along with the basic transformation in one place. Specifically, the equation for \bar{x}_i, s_i on top of page 3, the formula for L on the bottom of page 4, and the description of the bootstrap method for 95% CI of LOA can be repeated in one table or algorithm description succinctly. This way the reader can quickly understand the complete method in one place.

Answer: We agree, and we have added a box with the full algorithm (Algorithm 1) to which we refer at the end of the methods section “The full algorithm for preparing an extended Bland-Altman plot is presented as Algorithm 1.”

Also, one would want to know, for figures 1 and 2, the values of s or sigma so the numerical values of the y-axis can be made sense of. I assumed this was 1, but it is disorienting to try to make assumptions while trying to understand someone's method. Similarly for completeness sake, the parameter values for Figure 3 can also be given as a table.

Answer: Thank you for this important point. We have added the parameters of the models to the figure legends of both Figure 1 (true sigma = 2) and Figure 3 (true sigma = \sqrt(40) for rater 2 in the last row, all other true sigma = 2).

Reviewer 3 Report

An extension of Bland-Altman plot for analyzing agreement of more than two raters

The article presents an innovative development of the Bland-Altman plot, that has a high degree of originality and very promising practical applications in the analysis of agreement between more than two raters.  

This highly welcome tool for graphically transmitting easy-to-understand information regarding the agreement of several raters has been presented in a logical, clear, and highly transparent way by the authors of this article.

By contrasting numerous graphical examples from both simulation and real-world data, as well as by offering the detailed but easy-to-follow mathematical context of the proposed Bland-Altman plot extension, the authors have done an excellent job in clearly and objectively presenting their proposed method to the scientific community.

The method offers many advantages, not only regarding the more intuitive way of extending the classical Bland-Altman plot (compared to other proposed methods that could account for more than two raters) but also by displaying the limits of agreement between these raters along with their 95% confidence boundaries, which does indeed (as suggested by the authors) increase the likelihood that future users of this proposed plot will increase the frequency of reporting these highly informative confidence boundaries, compared to the current practice when using classic Bland-Altman plots.

The authors also clearly outline the strengths, limitations and future perspectives of their work, they use very clear English, and their conclusions seem well-tied and justified by the results of their scientific work.

Therefore, this reviewer wishes to congratulate the authors of this article for their excellent idea and its implementation, hereby recommending this article to be accepted for publication in the dedicated supplement issue of Diagnostics.

Author Response

An extension of Bland-Altman plot for analyzing agreement of more than two raters

The article presents an innovative development of the Bland-Altman plot, that has a high degree of originality and very promising practical applications in the analysis of agreement between more than two raters.  

This highly welcome tool for graphically transmitting easy-to-understand information regarding the agreement of several raters has been presented in a logical, clear, and highly transparent way by the authors of this article.

By contrasting numerous graphical examples from both simulation and real-world data, as well as by offering the detailed but easy-to-follow mathematical context of the proposed Bland-Altman plot extension, the authors have done an excellent job in clearly and objectively presenting their proposed method to the scientific community.

The method offers many advantages, not only regarding the more intuitive way of extending the classical Bland-Altman plot (compared to other proposed methods that could account for more than two raters) but also by displaying the limits of agreement between these raters along with their 95% confidence boundaries, which does indeed (as suggested by the authors) increase the likelihood that future users of this proposed plot will increase the frequency of reporting these highly informative confidence boundaries, compared to the current practice when using classic Bland-Altman plots.

The authors also clearly outline the strengths, limitations and future perspectives of their work, they use very clear English, and their conclusions seem well-tied and justified by the results of their scientific work.

Therefore, this reviewer wishes to congratulate the authors of this article for their excellent idea and its implementation, hereby recommending this article to be accepted for publication in the dedicated supplement issue of Diagnostics.

Answer: Thank you for the friendly review.

Reviewer 4 Report

The authors suggest a way to extend the Bland & Altman methodology to the setting with more than two raters.

General comments: I have major concerns with the proposed methodology to assess agreement between raters, as basically the proposed approach is very similar to the Bland & Altman methodology. Apparently, the authors are not aware of recent literature demonstrating that the limits of agreement (LoA) method is fundamentally flawed by an endogeneity bias, and the same criticism holds for the proposed methodology (Cartensen, 2010; Taffé, 2018; and other references provided below).

In Taffé (2018), it has been shown that the conditions for the LoA method to provide unbiased estimates are very special and unlikely to hold in practice.

It should also be mentioned that not only does the computation of the proposed LoA rely essentially on the normal distribution of the measured characteristic (it breaks down without normality…), but also their argument regarding the interpretation of the Y-axis does not hold for the case of >2 raters.

References

  • Carstensen B. Comparing methods of measurement: Extending the LoA by regression. Stat Med 2010, 29: 401-410.
  • Nawarathna LS, Choudhary PK. Measuring agreement in method comparison studies with heteroscedastic measurements. Stat Med 2013; 32: 5156-5171.
  • Nawarathna LS, Choudhary PK. A heteroscedastic measurement error model for method comparison data with replicate measurements. Stat Med 2015; 34: 1242-1258.
  • Taffé P. Effective plots to assess bias and precision in method comparison studies. Stat Methods Med Res 2018; 27: 1650-1660.

Author Response

The authors suggest a way to extend the Bland & Altman methodology to the setting with more than two raters.

Answer: Thank you for the interesting review, which points to an important limitation of our proposed method. We have amended the manuscript in order to discuss this issue, and we have answered your specific points below.

General comments: I have major concerns with the proposed methodology to assess agreement between raters, as basically the proposed approach is very similar to the Bland & Altman methodology. Apparently, the authors are not aware of recent literature demonstrating that the limits of agreement (LoA) method is fundamentally flawed by an endogeneity bias, and the same criticism holds for the proposed methodology (Cartensen, 2010; Taffé, 2018; and other references provided below).

In Taffé (2018), it has been shown that the conditions for the LoA method to provide unbiased estimates are very special and unlikely to hold in practice.

Answer: We have added respective references and acknowledged this important point both in the introduction and the discussion:

“While the main aim of this proposal is the comparison of multiple raters, the plot can be applied to comparisons of multiple methods as well. Especially in this case, it is important to be aware of the assumption of homoscedasticity, which our proposal inherits from the classic Bland-Altman plot; deviations from this assumption damage the interpretation of the Bland-Altman limits of agreement [Taffe2018, Carstensen2010b]”

 “An additional weakness of the proposed method is that it shares the assumption of homoscedasticity with the classical BA plot. It is well-known from literature that this assumption can be problematic, especially in the case of method comparison studies [Taffe2018, Carstensen2010b]. There have been detailed proposals of replacements for the classical BA plot, which handle heteroscedasticity in a more suitable manner both theoretically ([Nawarathna2013, Nawarathna2015]) and with practical applications and software implementations [Taffe2019, Taffe2017, Taffe2020]. We consider this problem to be mainly prominent in the case of method comparison studies and to a lesser degree when comparing raters. As the suggested approach by Taffé (similarly to the classical BA plot) only handles comparison of two methods, we consider our proposal to be a relevant extension of the classical BA plot in the case of multiple raters and homoscedasticity.”

It should also be mentioned that not only does the computation of the proposed LoA rely essentially on the normal distribution of the true latent trait, but their argument regarding the interpretation of the Y-axis does not hold for the case of >2 raters.

Answer: We agree that our explanation of how to interpret the y-axis was unclear. We have extended our elaboration on this:

“Investigating agreement between methods or raters, this LOA then can be compared to a clinically relevant difference by rescaling the y-axis with a factor of sqrt(2), which enables direct comparison of deviations from the different ratings as used in our method with differences of two ratings as used in classical BA plots. We have decided not to propose this rescaling in general for the extended plots, as the rescaled y-axis loses its clear statistical interpretation as an within-subject standard deviation.”

References

  • Carstensen B. Comparing methods of measurement: Extending the LoA by regression. Stat Med 2010, 29: 401-410.
  • Nawarathna LS, Choudhary PK. Measuring agreement in method comparison studies with heteroscedastic measurements. Stat Med 2013; 32: 5156-5171.
  • Nawarathna LS, Choudhary PK. A heteroscedastic measurement error model for method comparison data with replicate measurements. Stat Med 2015; 34: 1242-1258.
  • 2016; 35: 2328-2358.
  • Taffé P, Peng M, Stagg V, Williamson T. Bias plot: a package to effective plots to assess bias and precision in method comparison studies. Stata J 2017;17:208e21.
  • Taffé P. Effective plots to assess bias and precision in method comparison studies. Stat Methods Med Res 2018; 27: 1650-1660.
  • Taffé P, Peng M, Stagg V, Williamson T. MethodCompare: an R package to assess bias and precision in method comparison studies. Stat Methods Med Res 2019; 28: 2557-2565.
  • Taffé P. Assessing bias, precision, and agreement in method comparison studies. Stat Methods Med Res 2020; 3: 778-796.
  • Taffé P, Halfon P, and Halfon M. A new statistical methodology to assess bias and precision. J Cin Epi 2020; 24: 1-7.

Answer: Thank you. We have added most of these references to our manuscript.